# Leveraging Fully-Observable Solutions for Improved Partially-Observable Offline Reinforcement Learning

**Chulabhaya Wijesundara**                                   *wijesundara.c@northeastern.edu*
*Khoury College of Computer Sciences*
*Northeastern University*
*STR*

**Andrea Baisero**                                          *baisero.a@northeastern.edu*
*Khoury College of Computer Sciences*
*Northeastern University*

**Gregory Castañón**                                        *gregory.castanon@str.us*
*STR*

**Alan Carlin**                                              *alan.carlin@str.us*
*STR*

**Robert Platt**                                            *r.platt@northeastern.edu*
*Khoury College of Computer Sciences*
*Northeastern University*

**Christopher Amato**                                       *c.amato@northeastern.edu*
*Khoury College of Computer Sciences*
*Northeastern University*

**Reviewed on OpenReview:** *https://openreview.net/forum?id=e9p4TDPy6A*

## Abstract

Offline reinforcement learning (RL) is a popular learning framework for control problems where online interactions with the environment are expensive, risky, or otherwise impractical. Existing offline RL methods commonly assume full observability of the state, and therefore there is a lack of offline RL methods that are specialized for the more general case of partially-observable control. To address this gap, we propose Cross-Observability Conservative Q-Learning (CO-CQL), an offline RL algorithm for partially-observable control that leverages fully-observable expert policies in an asymmetric learning setting. To motivate the use of fully-observable experts for partially-observable control, we formalize Cross-Observability Optimality Ratio (COOR), a theoretical measure of cross-observability that quantifies the benefit of learning asymmetrically from a fully-observable expert, and Cross-Observability Approximation Ratio (COAR), an estimation of COOR computable from trained policies. Our empirical evaluation on a wide variety of partially-observable challenges demonstrates that CO-CQL is able to exploit the guidance of fully-observable experts to outperform other state-of-the-art offline algorithms.

## 1 Introduction

Standard *online* reinforcement learning (RL) involves learning from interactions with the environment (Sutton & Barto, 2018). However, many real-world control problems are inconducive to online interactions due to high costs and risks (Levine et al., 2020), e.g., autonomous driving and healthcare. To address this issue, *offline* RL (a.k.a *batch* RL (Fujimoto et al., 2019)) has emerged as a data-driven solution where agents learn

from pre-collected interactions (Levine et al., 2020). Many such approaches (Kumar et al., 2020; Kostrikov et al., 2021; Fujimoto & Gu, 2021) are developed assuming fully-observable (FO) control, whereas real-world control problems are often partially-observable (PO), featuring noisy observations and hidden information.

In this work, we address offline partially observable RL through the lens of *asymmetric* RL, where the learning agent can take advantage of privileged state information (Pinto et al., 2018; Baisero & Amato, 2022; Baisero et al., 2022) and/or a fully-observable expert policy (Weihs et al., 2021; Warrington et al., 2021; Nguyen et al., 2022; Shenfeld et al., 2023) during training. Though the use of state information during deployment is in clear conflict with the premise of partial observability, the availability of such information during training is possible and beneficial in a number of realistic scenarios, e.g., when training in simulation (via access to the simulation state), and when training in a highly controlled environment for deployment in a less controlled one (via the placement of additional sensors in the training environment).

Our asymmetric partially-observable offline RL setting assumes access to a dataset containing both states and observations, and to a fully-observable expert policy (which can itself be trained using the same dataset). Given these prerequisites, our method learns a partially-observable policy entirely offline without requiring online interactions. This reflects settings where access to a simulated or highly controlled training environment is available but expensive or limited, and where deployment must rely solely on partial observations, such as autonomous driving (Chen et al., 2019; Codevilla et al., 2018; Ge et al., 2020), robotic locomotion (Kumar et al., 2021; Margolis et al., 2022), and robotic grasping (Pinto et al., 2018; Chen et al., 2023). This setting captures a broad range of practical cases: (a) when both a simulator and fully-observable expert are available, (b) when a dataset is pre-collected and only expert supervision is needed, and (c) when the fully-observable expert is handcrafted or scripted and the simulator is only used to log data. Our method applies in all such scenarios, and furthermore, although rare, our method can also clearly be applied in the setting when both a mixed-observability dataset and fully-observable expert are available without requiring a simulator.

Our contributions in this work are summarized as follows: (a) We motivate asymmetric learning of partially-observable tasks from fully-observable experts, and propose *Cross-Observability Optimality Ratio* (COOR), as a theoretical measure to quantify the utility of fully-observable experts for partially-observable tasks; (b) We propose *Cross-Observability Approximate Ratio* (COAR) as a practical estimation of COOR; (c) We develop *Cross-Observability Conservative Q-Learning* (CO-CQL), a new offline RL algorithm that exploits asymmetric learning for partially-observable control by combining conservative Q-function regularization and behavior cloning objectives; (d) We perform an empirical evaluation of CO-CQL and other competitive baselines on a wide variety of identified partially-observable control task archetypes; and (e) We perform an analysis of robustness to dataset quality for CO-CQL. Our results demonstrate that in environments with high COOR, even simple behavior cloning from fully-observable experts is able to achieve good performances. Our hybrid CO-CQL method can leverage this guidance in high-COOR environments while improving upon it by optimizing the standard RL objective. In environments with low COOR, CO-CQL is still able to exploit the fully-observable guidance when useful, and ignore it when not.

## 2 Related Work

We briefly summarize other related work in offline RL and partially-observable control.

**Offline Fully-Observable RL.** Behavior Cloning (BC) (Bain & Sammut, 1999) employs supervised learning to learn directly from a dataset of expert demonstrations. Twin-Delayed DDPG with Behavior Cloning (TD3+BC) (Fujimoto & Gu, 2021) uses BC as an auxiliary objective on a policy primarily trained via TD3 (Fujimoto et al., 2018). Conservative Q-Learning (CQL) (Kumar et al., 2020) applies a conservative regularizer on value models to reduce overestimation biases. While CQL and TD3+BC aim to minimize the sampling of out-of-behavior-distribution actions, Implicit Q-Learning (IQL) (Kostrikov et al., 2021) avoids the issue by learning a value function from the dataset's state-action pairs. All of these methods are primarily designed and evaluated on fully-observable control, whereas we address partially-observable control.

**Online Partially-Observable RL.** Although many works have focused on online fully-observable RL, far fewer address partially-observable control (Weihs et al., 2021; Warrington et al., 2021). Cross-Observability Soft Imitation Learning (COSIL) (Nguyen et al., 2022) is the most similar to our work in using a fully-observable expert as part of an auxiliary behavior cloning objective. The key difference is that COSIL is an online algorithm, whereas ours is fully offline, with no environment feedback during learning. Furthermore, our algorithm requires key modifications to better fit the offline setting, e.g., additional regularization of the Q-function to reduce distribution shift and Q-value overestimation.

**Offline Partially-Observable RL.** Offline partially-observable RL is a challenging and under-explored area, as it inherits difficulties from both offline RL and partially-observable RL. Guo et al. (2022) recently proposed a theoretical approach to solving linear POMDPs via causal inference in a provably efficient manner. Concurrently, Lu et al. (2022) introduced Proxy variable Pessimistic Policy Optimization (P3O) which also uses causal inference for estimating pessimistic value functions. However, both works lack empirical evaluation, and primarily demonstrate theoretical advances to the field. In contrast, our proposed CO-CQL method is a practical algorithm that we are able to evaluate empirically. Gu et al. (2023) and Hong et al. (2023) explore representation learning in the offline setting. The former learns discrete proxy representations of *states*, which are then mapped to observation histories through a secondary encoder; a policy is learned to predict actions based on these proxy representations. The latter shows that combining a bisimulation metric with an offline algorithm can allow for learning of more compact *observation history* representations, thereby improving overall sample efficiency. Instead of high-level observation history abstraction, we focus on whether low-level MDP expert supervision can improve partially-observable offline learning. We hypothesize that the representation learning techniques of Gu et al. (2023) and Hong et al. (2023) could be integrated into our approach in future work to improve sample efficiency and performance.

## 3 Background

In this section, we briefly summarize the background in partially-observable control relevant to our work.

**Partially-Observable Markov Decision Processes.** A partially-observable Markov decision process (POMDP) (Åström, 1965) is a discrete-time control problem defined by a tuple $(\mathcal{S}, \mathcal{A}, \mathcal{O}, p_0, T, O, R, \gamma)$, with state space $\mathcal{S}$, action space $\mathcal{A}$, observation space $\mathcal{O}$, starting distribution $p_0(s)$, stochastic transition function $T(s, a, s')$, stochastic observation function $O(a, s, o)$, reward function $R(s, a)$, and discount factor $\gamma$.

Contrary to fully-observable agents, a partially-observable agent observes the state indirectly via noisy and stochastic observations. As a result, partially-observable policies act based on their observable action-observation history $h_t = (o_0, a_0, \ldots, o_{t-1}, a_{t-1}, o_t) \in \mathcal{H}$. The objective of partially-observable RL is to train a history-based policy $\pi \colon \mathcal{H} \to \Delta\mathcal{A}$ that maximizes the total expected episodic return $J_{\mathrm{RL}} = \mathbb{E}\left[\sum_t \gamma^t R(s_t, a_t)\right]$.

**Offline Reinforcement Learning.** Offline RL is a form of off-policy RL. Assuming fully-observable control, the agent is trained on a static dataset of fully-observable transitions $\mathcal{D}_{\mathrm{MDP}} = \left\{(s, a, r, s')_i\right\}_{i=1}^N$ generated by a behavior policy $\pi_\mathcal{B}$. The goal of offline RL is to train a policy $\pi$ to maximize the RL objective $J_{\mathrm{RL}}$ using the offline dataset rather than online interactions. In partially-observable control, the dataset contains observation data (typically in history form) rather than state data $\mathcal{D}_{\mathrm{POMDP}} = \left\{(h, a, r, o)_i\right\}_{i=1}^N$.

Learning a partially-observable policy solely on observation data is extremely difficult. Our work employs an asymmetric framework to aid training by assuming access to an offline dataset that includes *both* observation and state data $\mathcal{D} = \left\{(h, s, a, r, s', o)_i\right\}_{i=1}^N$, and a well-performing fully-observable expert $\mu \colon \mathcal{S} \to \Delta\mathcal{A}$.

**Conservative Q-Learning (CQL).** Kumar et al. (2020) develop a state-of-the-art offline RL algorithm for fully-observable control that employs the principle of pessimism (Buckman et al., 2020) to prevent value overestimation. In their work, Kumar et al. (2020) propose two forms for CQL. One form of CQL combines a standard DQN objective (Mnih et al., 2015)

$$J_{\mathrm{CQL}}^Q = \frac{1}{2} \mathbb{E}_{s,a,r,s' \sim \mathcal{D}} \left[ \left( r + \gamma \max_{a'} Q(s', a') - Q(s, a) \right)^2 \right] + \lambda \mathcal{R}(Q), \tag{1}$$

with a conservative value regularizer, $\mathcal{R}(Q) = \mathbb{E}_{s\sim\mathcal{D}}\left[\max_a Q(s,a)\right] - \mathbb{E}_{s,a\sim\mathcal{D}}\left[Q(s,a)\right]$ that minimizes the gap between maximal and in-distribution values. When the action space is continuous, an auxiliary policy model $\mu(s) \approx \text{argmax}_a Q(s,a)$ is trained to estimate maximal values.

Another form of CQL combines soft actor-critic (SAC) objectives with the value regularizer,

$$J_{\text{CQL}}^{\pi} = \mathbb{E}_{s\sim\mathcal{D},a\sim\pi(s)}\left[\alpha \log \pi(a \mid s) - Q(s,a)\right], \tag{2}$$

$$J_{\text{CQL}}^{Q} = \frac{1}{2}\,\mathbb{E}_{s,a,r,s'\sim\mathcal{D}}\left[\left(r + \gamma\,\mathbb{E}_{a'\sim\pi(s')}\left[Q(s',a') - \alpha \log \pi(a' \mid s')\right] - Q(s,a)\right)^2\right] + \lambda\mathcal{R}(Q). \tag{3}$$

**Cross-Observability Soft Imitation Learning (COSIL).** Nguyen et al. (2022) propose an online RL method that exploits a pre-trained fully-observable agent to guide the training of a partially-observable agent via an objective that incorporates a policy divergence $D$ as a pseudo-reward,

$$J_{\text{COSIL}} = \mathbb{E}\left[\sum_t \gamma^t \left(R(s_t, a_t) - \alpha D\left(\mu(s_t), \pi(h_t)\right)\right)\right], \tag{4}$$

## 4 Leveraging Fully-Observable Solutions for Partially-Observable Offline RL

In this section we first introduce a theoretical framework to justify learning via cross-observability, then a practical approximation to it that is computable, and finally CO-CQL, our proposed algorithm based on conservative cross-observability imitation learning for offline RL.

### 4.1 Cross-Observability Optimality Ratios

Let $\mathcal{A}_{\text{PO}}^*(h), \mathcal{A}_{\text{FO}}^*(s) \subseteq \mathcal{A}$ denote the subsets of actions that are optimal for a partially-observable agent and a fully-observable agent respectively acting on a history $h$ and a state $s$. It is a fundamental property of partially-observable control that $\mathcal{A}_{\text{PO}}^*(h)$ and $\mathcal{A}_{\text{FO}}^*(s)$ are not generally equivalent, and agents acting upon different information may have to take different actions to act optimally. However, this is not necessarily always the case, and many control problems feature a significant overlap between optimal partially-observable and fully-observable actions. We quantify this notion as a *cross-observability optimality ratio*.

**Definition 4.1** (Cross-Observability Optimality Ratio (COOR)). For any given history and state, $\rho^*(h,s) \in [0,1]$ is the ratio of optimal fully-observable actions that are also optimal for partially-observable control.

$$\rho^*(h,s) = \frac{|\mathcal{A}_{\text{PO}}^*(h)\bigcap \mathcal{A}_{\text{FO}}^*(s)|}{|\mathcal{A}_{\text{FO}}^*(s)|}. \tag{5}$$

**Simplified HeavenHell Example.** Consider the simplified variant of *HeavenHell* (Blai & Geffner, 1998) shown in Figure 1. The agent must identify and reach the *good* exit while avoiding the *bad* exit. A fully-observable agent directly observes the *good* exit location, and moves accordingly. A partially-observable agent must first visit an *oracle* to identify the exits and reduce its state uncertainty, and then backtrack to reach the *good* exit. Although the optimal partially-observable policy and the optimal fully-observable policy differ, there are several history-state pairs (about two thirds) with overlapping optimal behaviors.

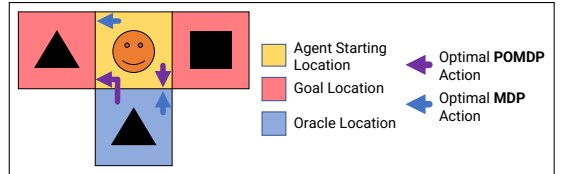

Figure 1: Simplified *HeavenHell*. A partially-observable agent must visit an *oracle* to gather information, while a fully-observable agent does not.

**Approximate Ratio.** Computing COOR requires expert domain knowledge of all optimal actions for both partially-observable and fully-observable agents—a clearly impractical assumption. To approximate COOR in practice, we propose a measure of the overlap between a *well-trained* partially-observable policy and an expert fully-observable policy.

**Definition 4.2** (Cross-Observability Approximation Ratio (COAR)). Let $\pi$ and $\mu$ be well-trained partially-observable and fully-observable policies respectively, and $q \in [0, 1]$ be a freely chosen probability threshold. Then, $\hat{\rho}(h, s) \in [0, 1]$ is the ratio of $q$-likely fully-observable actions that are also $q$-likely according to the partially-observable policy,

$$\hat{\rho}(h, s) = \frac{\sum_a \mathbb{I}\left[\pi(a \mid h) \geq q\right] \cdot \mathbb{I}\left[\mu(a \mid s) \geq q\right]}{\sum_a \mathbb{I}\left[\mu(a \mid s) \geq q\right]}. \tag{6}$$

In practice, each indicator is an estimate of the optimal subsets $\mathcal{A}_{\text{PO}}^*(h)$ and $\mathcal{A}_{\text{FO}}^*(s)$. Examining the performance of a trained partially-observable policy in conjunction with the corresponding COAR, we can estimate the task cross-observability. We will perform such an analysis in our evaluation.

**On the role of COOR.** COOR is not used directly in our algorithm, and we do not derive bounds based on it. Instead, it serves a motivational and analytical role: it formalizes the intuition that some partially-observable control problems share significant overlap with their fully-observable counterparts, and that leveraging fully-observable guidance can be beneficial. We show empirically in Section 5.2.5 that COAR correlates strongly with the effectiveness of behavior cloning from fully-observable experts. This validates both COOR and COAR as useful tools for interpreting the success of asymmetric offline learning.

### 4.2 Cross-Observability Conservative Q-Learning for Partially-Observable Offline RL

CO-CQL makes several extensions to CQL and COSIL: (a) To adequately handle the partially-observable data, state-based models are replaced with history-based models (e.g., $\pi(s)$ becomes $\pi(h)$, $Q(s, a)$ becomes $Q(h, a)$), using recurrent networks to process sequential data; (b) To exploit the fully-observable expert, we add a BC auxiliary loss to the policy objective; and (c) To handle discrete control problems, we employ the specialized discrete implementation of SAC (Christodoulou, 2019).

We adapt conservative regularization $\mathcal{R}(Q) = \mathbb{E}_{h \sim \mathcal{D}}\left[\max_a Q(h, a)\right] - \mathbb{E}_{(h,a) \sim \mathcal{D}}\left[Q(h, a)\right]$ to partially-observable control. As in SAC, we use this in a value objective adapted to offline partially-observable control,

$$y = r + \gamma \mathbb{E}_{a' \sim \pi(hao)}\left[Q(hao, a') - \alpha \log \pi(a' \mid hao)\right], \tag{7}$$

$$J_{\text{CO-CQL}}^Q = \frac{1}{2} \mathbb{E}_{h,a,r,o \sim \mathcal{D}}\left[(y - Q(h, a))^2\right] + \lambda \mathcal{R}(Q). \tag{8}$$

where $hao$ denotes the extended observation history: the current history $h$, followed by action $a$, and the resulting new observation $o$.

The entropy temperature objective of SAC is likewise adapted to offline RL via

$$J_{\text{CO-CQL}}^\alpha = \alpha \mathbb{E}_{h \sim \mathcal{D}, a \sim \pi(h)}\left[-\log \pi(a \mid h)\right] - H, \tag{9}$$

where $H$ is the target entropy value. Lastly, the policy objective of SAC is adapted to the offline partially-observable control and combined with an auxiliary BC term based on imitating the fully-observable expert,

$$J_{\text{CO-CQL}}^\pi = \mathbb{E}_{h,s \sim \mathcal{D}}\left[\mathbb{E}_{a \sim \pi(h)}\left[\alpha \log \pi(a \mid h) - Q(h, a)\right] + \beta D\left(\mu(s) \mid\mid \pi(h)\right)\right], \tag{10}$$

where $\beta$ is a BC scaling coefficient and $D$ is a divergence measure between fully-observable and partially-observable policies whose form depends on the action space.

We interpret this behavior cloning term as imitation learning that tries to project fully-observable behavior into partially-observable behavior space. In an online setting such as that of COSIL (Nguyen et al., 2022), the behavior cloning term is useful to reduce exploration time, as the fully-observable expert has already done some amount of exploration during its own training. Because this notion of exploration does not exist in the offline setting, CO-CQL focuses on exploiting the recommendations of the expert policy to resolve blind spots that would result from a partially-observable policy being trained exclusively on the dataset.

**Continuous Control.** When the action space is continuous, we approximate the expectations over actions in Equations (7), (9) and (10) via Monte Carlo (MC) estimation. Further, we employ an expected mean squared difference error $D(\mu(s) \mid\mid \pi(h)) = \mathbb{E}_{u \sim \mu(s), a \sim \pi(h)}\left[\|u - a\|_2^2\right]$ as the divergence measure between fully-observable and partially-observable policies (again estimated via MC).

**Discrete Control.** When the action space is discrete, the expectations over actions in Equations (7), (9) and (10) are computed exactly via full enumeration, i.e.,

$$\mathbb{E}_{a' \sim \pi(hao)} \left[ Q(hao, a') - \alpha \log \pi(a' \mid hao) \right] = \sum_{a'} \pi(a' \mid hao) \left( Q(hao, a') - \alpha \log \pi(a' \mid hao) \right), \qquad (11)$$

$$\mathbb{E}_{a \sim \pi(h)} \left[ -\log \pi(a \mid h) \right] = -\sum_a \pi(a \mid h) \log \pi(a \mid h). \qquad (12)$$

Further, we employ cross-entropy $D(\mu(s) \mid\mid \pi(h)) = -\sum_a \mu(a \mid s) \log \pi(a \mid h)$ as a divergence measure.

**Caveats on Context-Independent Behavior Cloning.** It is clear that different history-state contexts imply varying degrees of cross-observability, as reflected by COOR $\rho^*(h, a)$. Ideally, we would want to regulate the amount $\beta$ of behavior cloning in a context-dependent fashion, e.g., based on COOR values. Unfortunately, COOR is currently a theoretical construct that is not available in practice, and even the corresponding COAR $\hat{\rho}(h, a)$ is only computable post-training. Though more sophisticated methods are likely desirable, we are currently limited to applying a single scalar factor $\beta$ chosen on the properties of the control problem as a whole. Finding better ways to regulate $\beta$ represents an important focus for future work.

## 5 Evaluation

In this section we provide an overview of our experimental setup. We perform two types of evaluations: an analysis of learning performance comparing CO-CQL to other baselines, and an analysis on how robust CO-CQL is to the quality of the training dataset. The Appendix contains more detailed discussions on control problems, datasets, architectures, and hyperparameters.

**Baselines.** We implement non-asymmetric baselines of direct behavior cloning of the fully-observable expert (BC), discrete and continuous variants of recurrent CQL and IQL, and a continuous variant of recurrent TD3+BC. During training CQL, IQL, and TD3+BC use observation histories, while asymmetric CO-CQL uses both observation histories and states.

**Evaluation Metrics.** Each method is evaluated over 5 independent learning runs. Each run is periodically evaluated by computing an average performance over 10 sample episodes. The trained policies of all methods only use observation histories during evaluation. For each method, we report the *mean* and *standard error* of these average performances, over the 5 runs.

### 5.1 Control Problems

We identify 4 prototypical challenges concerning partial observability. Each challenge is evaluated on a corresponding dataset as described in Appendix A.6:

**Noisy Observations** The agent observes a noisy version of the state corrupted by Gaussian noise. We employ modified versions of *HalfCheetah* and *LunarLander* (Brockman et al., 2016).

**Latent State Dimensions** The agent consistently observes some but not all state dimensions. We employ modified versions of *HalfCheetah* and *LunarLander* (Brockman et al., 2016) with observations that maintain position information but lack velocity information.

**Information-Gathering** The agent must first obtain key information, and then rely on long-term memorization to remember it and complete the task. We employ a hard variant of *HeavenHell* (Blai & Geffner, 1998) and *MemoryFourRooms* (Baisero et al., 2021).

**Field of View** The agent observes its environment via a limited field of view. This is similar to latent state dimensions, with the difference that the dimensions that remain latent change based on the agent's behavior. We employ *DynamicObstacles* and *KeyDoor* (Baisero et al., 2021).

In addition to the above, we perform an analysis on dataset robustness for CO-CQL based on modified versions of *CartPole* (Brockman et al., 2016) and simplified *HeavenHell* (Blai & Geffner, 1998).

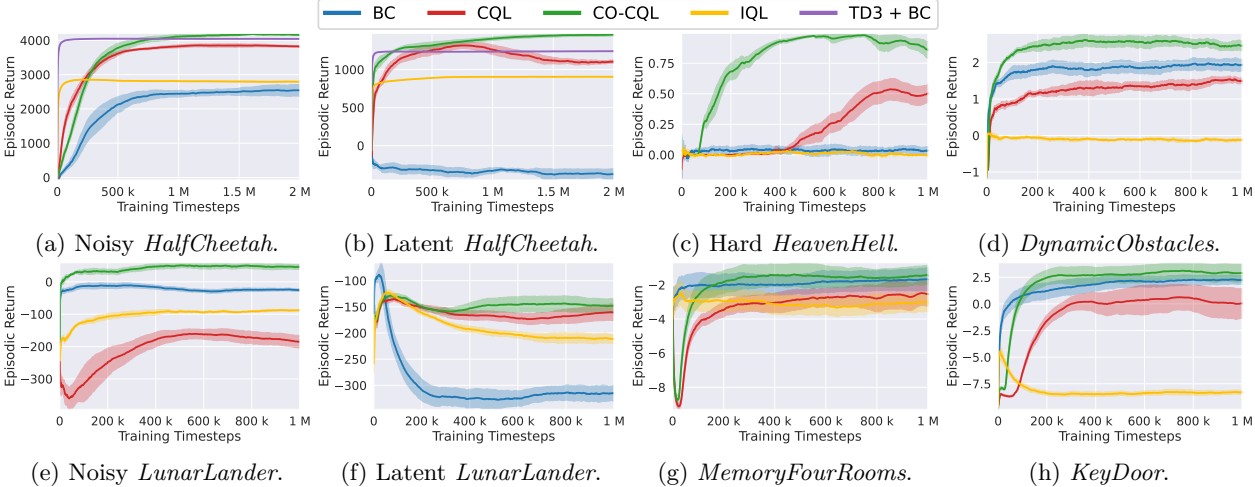

Figure 2: Performance mean and standard error (5 runs).

## 5.2 Results

Sections 5.2.1 to 5.2.4 contains a performance analysis of evaluated methods on the 4 partially-observable challenges. Section 5.2.5 contains an analysis on the behavior cloning coefficient $\beta$ of CO-CQL and its relationship to COOR. Section 5.2.6 contains an analysis of robustness to dataset quality for CO-CQL.

### 5.2.1 Noisy Observation Analysis

Noisy observations require the agent to learn to filter the noise over multiple observations in order to recover a better estimation of the underlying state. The usefulness of behavior cloning form a fully-observable expert likely depends on the amount of noise injected relative to the complexity of the underlying state dynamics, with less noise and simpler dynamics making behavior cloning more directly useful, whereas more noise and more complex dynamics make behavior cloning less useful.

**Noisy *HalfCheetah*.** This task requires precise control of tightly coupled joints, and we expect the addition of noise to have a major effect on optimal control. Consequently, we expect BC to struggle. CO-CQL performs the best, followed closely by TD3+BC and CQL. We hypothesize that (unlike IQL and naive BC), CO-CQL, TD3+BC, and CQL are helped in the partially-observable setting by the use of conservative regularization. Additionally for CO-CQL, the behavior cloning regularizer provides a stabilizing signal while learning on noisy observations. This indicates the usefulness of our behavior cloning regularizer in its ability to exploit even lower amounts of overlap. In Figure 2a, we see that indeed BC performs the worst.

**Noisy *LunarLander*.** In contrast to the above, the dynamics of this task are likely simple enough where we expect BC to perform well despite the introduction of observation noise. In Figure 2e, we see that indeed BC performs almost the best (behind CO-CQL). CO-CQL is able to exploit its BC component in this environment, leading to the best performance. Other baselines struggle to learn from the noisy experiences alone, demonstrating the benefits of exploiting existing cross-observability overlaps.

### 5.2.2 Latent State Dimensions Analysis

In principle, the latent state fields of these tasks are inferrable by integrating historic observations, e.g., a joint angular velocity can be estimated well by extrapolating observed angular positions. Instead of observing this information directly, it is the history model that is now burdened with the additional task to learn to perform this inference. Consequently, we expect high COOR and good performance by BC. However, Figures 2b and 2f show that this is not the case, indicating that the history models fail to appropriately estimate the missing state dimensions when guided by BC alone.

Figure 3: Hard *HeavenHell* visitation densities for the fully-observable expert (**left**), CQL (**center**), and CO-CQL (**right**), calculated across 100 episodes where the *good* exit is left. The fully-observable expert ignores the *oracle* altogether, introducing issues in a pure BC-based learning. CQL succeeds in visiting the *priest* but then fails to exploit the information gained. CO-CQL succeeds at balancing both sub-goals.

**Latent *HalfCheetah*.** Although BC performs poorly, we find that CO-CQL is still able to exploit the fully-observable guidance and perform better than other baselines. This plausibly indicates that the RL objective is useful to learn useful history representations, given which fully-observable guidance can be exploited.

**Latent *LunarLander*.** Again, we note that CO-CQL performs much better than BC, indicating that the fully-observable guidance is not sufficient to learn an appropriate history model. In this task with simpler dynamics, we note that CO-CQL and CQL perform similarly, with CO-CQL taking a small lead.

### 5.2.3 Information-Gathering Analysis

In problems that require information gathering and memorization, purely cloning the fully-observable expert results in suboptimal partially-observable behaviors that are unable to gather information. In partially-observable navigation tasks, this means that the partially-observable policy trained on BC alone can at best guess the correct navigation goal with a 50 % chance of success. Figure 3 shows *HeavenHell* visitation densities for the fully-observable expert, CQL, and CO-CQL, demonstrating that a fully-observable expert cannot demonstrate information gathering experience, but can still be used by CO-CQL to learn how to exploit the information after it has been gathered.

**Hard *HeavenHell*.** Information-gathering plays a fundamental role in this task, and we expect to see a stark difference in performance, depending on whether fully-observable guidance is used and whether it is used adequately. In Figure 2c, we see that most methods fail to perform well. BC fails because it has no guidance on how to visit the *oracle*, whereas CQL and IQL fail to solve the task as a whole. Figure 3 provides a visual and numerical breakdown of learned policy state visitation behavior across the fully-observable expert, CQL, and CO-CQL. The fully-observable expert, which also represents the policy learned through simple behavior cloning, goes directly to an exit while not performing any information-gathering behavior (as shown in the figure, the *oracle* location is visited 0% of the time). Standard CQL on the other hand, learns the opposite extreme of behavior, where it excessively visits the *oracle* and rarely follows through to the *good* exit. In contrast to both of these approaches, CO-CQL is able to exploit the fully-observable guidance of how to reach the *good* exit, while also learning to gain the key piece of information by visiting the *oracle*. As shown in the figure, the state distribution for the CO-CQL agent is spread more evenly, particularly with higher visitation for the starting state which is expected for an agent that needs to visit the *oracle* and then double back through the starting state to reach an exit.

***MemoryFourRooms*.** In this task, BC goes back to outperforming IQL and CQL. This is likely because it is possible that the agent observes the *beacon* (the key piece of information) on its way to the *good* exit, which was not possible in *HeavenHell*. Hence, the guidance of BC is at least sometimes more suitable even for partially-observable control. CO-CQL is able to exploit this guidance to perform significantly better than either IQL and CQL, and slightly better than BC.

### 5.2.4 Field of View Analysis

Both tasks are goal-reaching tasks performed from the agent's first-person point of view. In such tasks, the partial observability limits the agent's view, meaning that the agent may have to explore the layout to

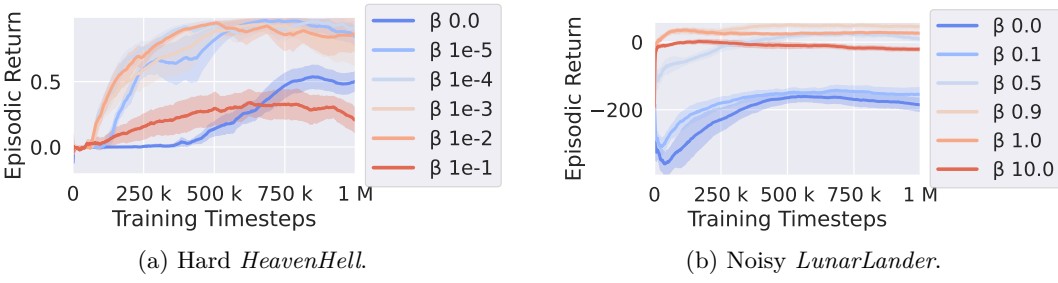

(a) Hard *HeavenHell.*

(b) Noisy *LunarLander.*

Figure 4: Performance mean and standard error (5 runs).

figure out the location of goals and subgoals; however, the information-gathering component of these tasks is comparatively low, as the agent is typically able to gain a broader view of the environment by moving or turning slightly. Consequently, we expect these environments to have high COOR, and fully-observable expert guidance to play an important role.

***DynamicObstacles.*** This task contains stochastically moving obstacles that need to be avoided while reaching a goal. Figure 2d confirms that indeed BC performs very well, better than methods like CQL that do not exploit fully-observable expert guidance. Nonetheless, CO-CQL is able to exploit this guidance and combine it with the main RL objective to achieve the best learning performance.

***KeyDoor.*** This task contains navigation subgoals (the *key* and the *door*) that must be reached before the final *exit*. Once again, Figure 2h confirms that BC performs very well, demonstrating the benefit of exploiting fully-observable expert guidance in environments with high COOR. Once again, CO-CQL is able to exploit this guidance to achieve the best learning performance.

### 5.2.5 Cross-Optimality Ratio Analysis

We analyze how the behavior cloning coefficient $\beta$ influences the COAR estimate, and how both relate to learning performance. Figure 4 shows the performance of CO-CQL on Hard *HeavenHell* and Noisy *LunarLander* for different values of $\beta$, while Table 1 contains the corresponding COAR estimates. In Hard *HeavenHell*, the distance to the *oracle* and the *good* exit are similar, resulting in a known COOR of about 66 %. In contrast, the COOR of Noisy *LunarLander* is not known exactly, but we can make an informed guess that it is higher, as the task does not require much information-gathering. r

Overall, the results in Figures 4a and 4b confirm that low COOR requires lower $\beta$.

Table 1: Average COAR (5 runs).

| (a) *HeavenHell* | | (b) *LunarLander* | |
|---|---|---|---|
| $\beta$ | **COAR** | $\beta$ | **COAR** |
| 0 | 42.2 % | 0.0 | 62.2 % |
| 1e−5 | 69.2 % | 0.1 | 69.9 % |
| 1e−4 | 62.6 % | 0.5 | 83.4 % |
| 1e−3 | 59.2 % | 0.9 | 87.6 % |
| 1e−2 | 76.7 % | 1.0 | 90.6 % |
| 1e−1 | 85.8 % | 10.0 | 75.9 % |

**Hard *HeavenHell.*** In this task, we see that both high behavior cloning ($\beta = 1e-1$) and low behavior cloning ($\beta = 0$) result in poor performance. This is expected for tasks where guidance is useful up to some amount, but too much guidance is detrimental to information-gathering. Cross-referencing Figure 4a with Table 1a, we note that high performing CO-CQL policies ($\beta \in \{1e-5, 1e-4, 1e-3\}$) are associated with a COAR of about 66 %, broadly matching the true COOR. This confirms that the overlap between the learned policy and the fully-observable expert matches the overlap between the optimal partially-observable policy and the fully-observable expert.

**Noisy *LunarLander.*** In Figure 4b, we notice that the higher COOR of this environment translates into higher values of $\beta$ performing better. Even in such high overlap scenarios, there is often benefit in avoiding too high values of $\beta$, which allows CO-CQL to learn from the fully-observable expert, but also adapt to the

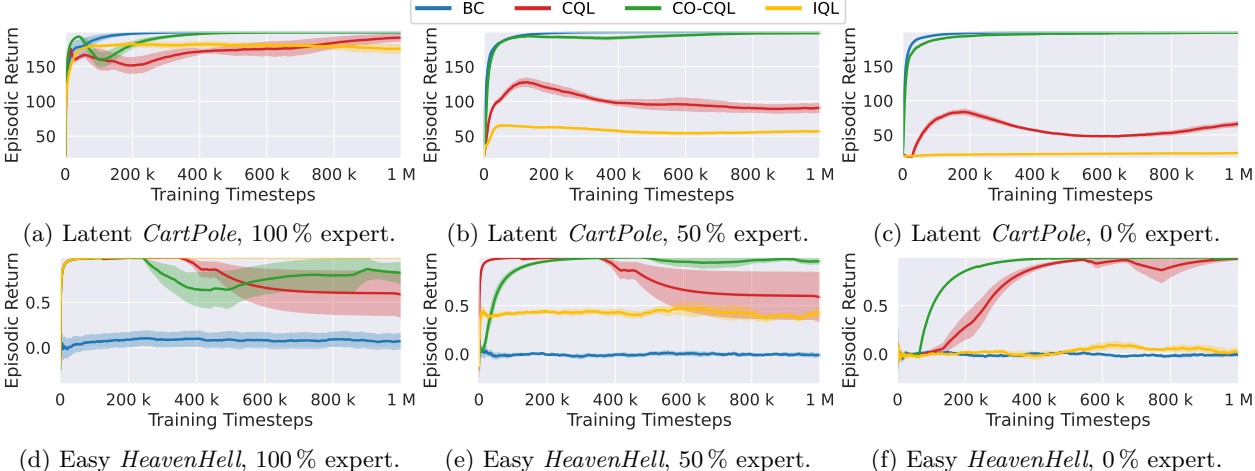

Figure 5: Performance mean and standard error (5 runs).

remaining partial observability. Nonetheless, CO-CQL demonstrates robustness towards $\beta$, as even extremely high values ($\beta = 10.0$) performs quite similar to the best value ($\beta = 1.0$).

### 5.2.6 Robustness to Dataset Quality Analysis

We evaluate the robustness of CO-CQL and baselines to dataset quality on Latent *CartPole* and Easy *HeavenHell*. For each control problem, we create datasets made up from partially-observable *expert* and *random* demonstrations in relative ratios of 100 %, 50 %, and 0 % respectively. Results are shown in Figure 5.

**Latent *CartPole*.** In this task, we expect partial observability to have minimal impact on the resulting optimal policy, and a high cross-optimality ratio. As expected, BC performs very well regardless of dataset quality. CO-CQL is able to exploit the high cross-optimality and achieve similar robustness across dataset qualities. In contrast, CQL and IQL are more susceptible to the dataset quality.

**Easy *HeavenHell*.** In this task, partial observability has a much bigger impact on the resulting optimality, and the cross-optimality is much more limited. As expected, BC fails regardless of the dataset quality, whereas the other more data-driven methods perform better. IQL performs well with high quality data, but is strongly affected by the quality as well. In contrast, CQL and CO-CQL perform well across all types of datasets. Of these two, CO-CQL exhibits the better performance due to its ability to exploit the fully-observable expert guidance when useful and ignore it when not.

## 6 Conclusion

In this work, we demonstrated that fully-observable experts can be used to aid the training of partially-observable agents in offline RL. We provide two novel measures of cross-observability: COOR, which is grounded by relevant theoretical insights, and COAR, which is a practical approximation useful to analyze the cross-observability properties associated with trained policies. Our algorithmic contribution, CO-CQL, combines a standard RL objective with an auxiliary behavior cloning objective, so as to effectively exploit both sources. Our empirical evaluation demonstrates that CO-CQL succeeds in exploiting fully-observable guidance in a diverse variety of partially-observable tasks and, critically, CO-CQL is able to learn well even when the task cross-observability is relatively low. Most importantly, CO-CQL is able to learn to solve sub-tasks that cannot be learned by the fully-observable expert alone.

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

# A    Appendix

## A.1    Low COOR Case Study

We consider a simplified version of the famous partially-observable Tiger environment (Kaelbling et al., 1998), where there is a 2-room grid with actions Listen, GoLeft, GoRight. The treasure is randomly placed left or right each episode. The fully-observable optimal policy always goes straight to the goal; the partially-observable optimal policy must first take Listen (reveals the treasure side) and then go to the revealed side. Before information is acquired, the optimal fully-observable action (go directly) is not optimal for the partially-observable policy; after listening, they align. Thus, across most (history, state) pairs before listening, the overlap of optimal actions is near zero, i.e., low COOR by construction, while it becomes high only after information acquisition. This illustrates why a strong BC term may be harmful in low-overlap phases and motivates smaller $\beta$ until overlap increases.

This simplified environment can be further extended to a zero COOR version, where the Listen action immediately ends the episode. In this variant, the optimal fully-observable policy always goes straight to the goal, and the optimal partially-observable policy always takes the Listen action to avoid receiving a penalty from blindly going to the wrong side. Thus, the overlap of optimal actions is zero. In this case, the $\beta$ BC coefficient should be set to zero to gracefully degrade CO-CQL to standard CQL because there is no MDP overlap to exploit.

## A.2    Compensating For The Loss Function

Because BC divergence is helpful primarily when fully-observable and partially-observable optima overlap, we hypothesize two approaches for compensating for low-overlap contexts. One is advantage gating, where the BC term is only included when the difference in Q-values for a dataset action vs. an MDP expert action

is larger than a specified threshold. Second, the BC term could be down-weighted based on the uncertainty of the actor policy, measured through entropy. In either case, CO-CQL could be set to fall back to use the dataset's actions rather than the MDP expert's actions. In this work we intentionally keep $\beta$ global for simplicity and validating the asymmetric learning approach and leave these extensions for future work.

## A.3   Environments

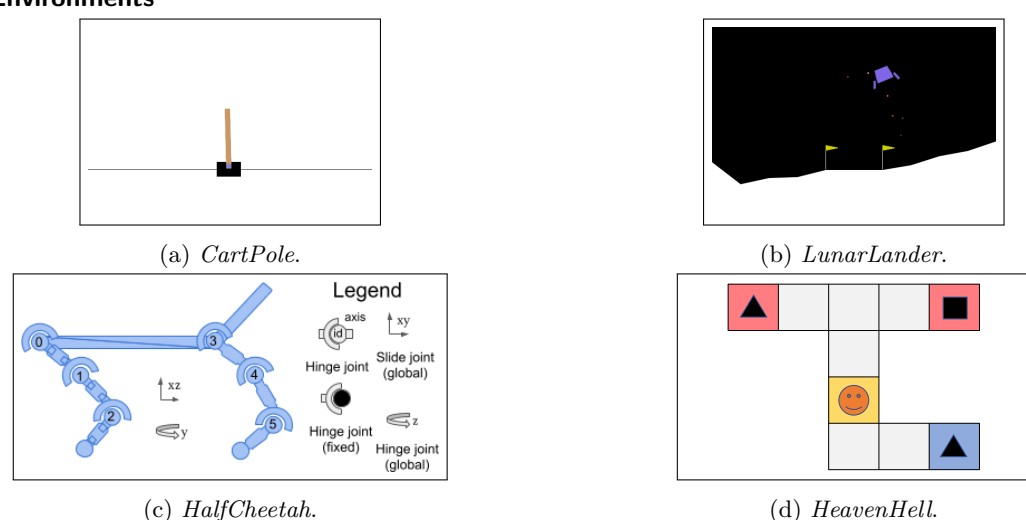

(a) *CartPole.*            (b) *LunarLander.*

(c) *HalfCheetah.*            (d) *HeavenHell.*

Figure 6: Environments.

This section contains detailed descriptions of the environments used in our evaluation.

### A.3.1   *CartPole*

*CartPole* (Barto et al., 1983) is a classic fully-observable control problem where the agent must learn to balance a pole on a moving cart. The agent receives a positive reward of 1 for each timestep that the pole does not fall. Its state space is continuous with $\dim(\mathcal{S}) = 4$ dimensions representing the position and velocity of the cart, and the angular position and velocity of the pole. Its action space is discrete with 2 actions, indicating whether the agent moves left or right. Figure 6a shows a rendering of *CartPole*.

**Latent *CartPole*.** This variant generates observations by removing velocity information from the state, reducing the observation dimensions $\dim(\mathcal{O})$ from 4 to 2.

### A.3.2   *LunarLander*

*LunarLander* (Brockman et al., 2016) is a classic fully-observable control problem where the agent must learn to pilot and land a spacecraft without crashing. The agent receives a mixture of sparse and dense rewards depending whether the agent landed safely or crashed, and how close it gets to the landing spot. Its state space is continuous with $\dim(\mathcal{S}) = 8$ dimensions denoting the 2D position, 2D velocity, 1D angular position, 1D angular velocity, and 2 binary dimensions representing whether either leg of the spacecraft is touching the ground. Its action space is discrete with 4 actions denoting any combination of left and right engine being on or off. Figure 6b shows a rendering of *LunarLander*.

**Noisy *LunarLander*.** This variant generates observations by adding Gaussian noise $\mathcal{N}(\mu = 0, \sigma^2 = 0.04)$ to the state.

**Latent *LunarLander*.** This variant generates observations by removing velocity information from the state, reducing the observation dimensions $\dim(\mathcal{O})$ from 8 to 5.

### A.3.3    *HalfCheetah*

*HalfCheetah* (Brockman et al., 2016) is a classic fully-observable control problem where the agent must learn to apply joint torques to make a cross-sectional robot run. The agent receives positive dense rewards for moving forward, and negative dense rewards for using large actions torques (to promote efficient movement). Its state space is continuous with $\dim(\mathcal{S}) = 17$ dimensions denoting the 6D angular positions of the controlled joints, the 6D angular velocities of the controlled joints, the 1D angular position of the uncontrolled head joint, the 1D angular velocity of the uncontrolled head joint, the 1D position of the head tip (only horizontal), and the 2D velocities of the head tip (horizontal and vertical). Its action space is continuous with $\dim(\mathcal{A}) = 6$ representing the 6D torques applied to the controlled joints. Figure 6c shows a rendering of *HalfCheetah*.

**Noisy *HalfCheetah*:** This variant generates observations by adding Gaussian noise $\mathcal{N}(\mu = 0, \sigma^2 = 0.04)$ to the state.

**Latent *HalfCheetah*:** This variant generates observations by removing velocity information from the state, reducing the observation dimensions $\dim(\mathcal{O})$ from 17 to 8.

### A.3.4    *HeavenHell*

*HeavenHell* (Blai & Geffner, 1998) is a classic partially-observable control problem where the agent must identify which of two exits is *good* and reach it. The agent receives a positive sparse reward of $+1$ only upon reaching the *good* exit, and a negative sparse reward of $-1$ only upon reaching the *bad* exit, regardless of whether it has acquired the appropriate information. Its state space is discrete, with a number of states that depends on the environment size, denoting the agent's position in the grid, and the location of the *good* exit. Its observation space is discrete, with a number of observations that depends on the environment size, denoting the agent's position, except when the agent is at the *oracle*, in which case the observation denotes the location of the *good* exit. Its action space is discrete with $\dim(\mathcal{A}) = 4$ representing the four directions where the agent can move (north, south, east, west).

**Easy *HeavenHell*:** This variant has 10 locations for the agent, and 2 possible locations for the *good* exit, adding up to $\dim(\mathcal{S}) = 10 \cdot 2 = 20$ states and $\dim(\mathcal{O}) = 10 - 1 + 2 = 11$ observations.

**Hard *HeavenHell*:** This variant has 14 locations for the agent, and 2 possible locations for the *good* exit, adding up to $\dim(\mathcal{S}) = 14 \cdot 2 = 28$ states and $\dim(\mathcal{O}) = 14 - 1 + 2 = 15$ observations.

### A.3.5    *GridVerse*

*GridVerse* (Baisero et al., 2021) is a framework for customizable 2D grid world control problems that allows for both full state observability and partial observability. In our evaluation, we use three predefined tasks: *DynamicObstacles*, *KeyDoor*, and *MemoryFourRooms*. For each of these tasks, we employ $7 \times 7$ states and $2 \times 3$ first-person POV observations.

The states of *GridVerse* environments are represented by the following components:

**Grid.** The grid field is a $7 \times 7 \times 3$ tensor of categorical data representing three channels of information for each of the $7 \times 7$ grid tiles. The first channel contains a categorical index representing the tile type, e.g., *wall*, *empty*, *exit*, *door*, *key*, etc. The second channel contains (when relevant) information on the state of the tile, e.g., whether a *door* tile is *open*, *closed*, or *locked*. The third channel contains (when relevant) color information on the object, which may be relevant for the dynamics of the environment, e.g., a *yellow door* may only be opened by a *yellow key* but not a *red key*.

**Agent ID Grid.** The agent ID grid field is a $7 \times 7$ binary matrix representing the agent's location.

**Agent** The agent field is a 6D array containing the pose of the agent, with 2 dimensions representing the agent coordinates, and 4 dimensions representing a one-hot encoding of agent orientation.

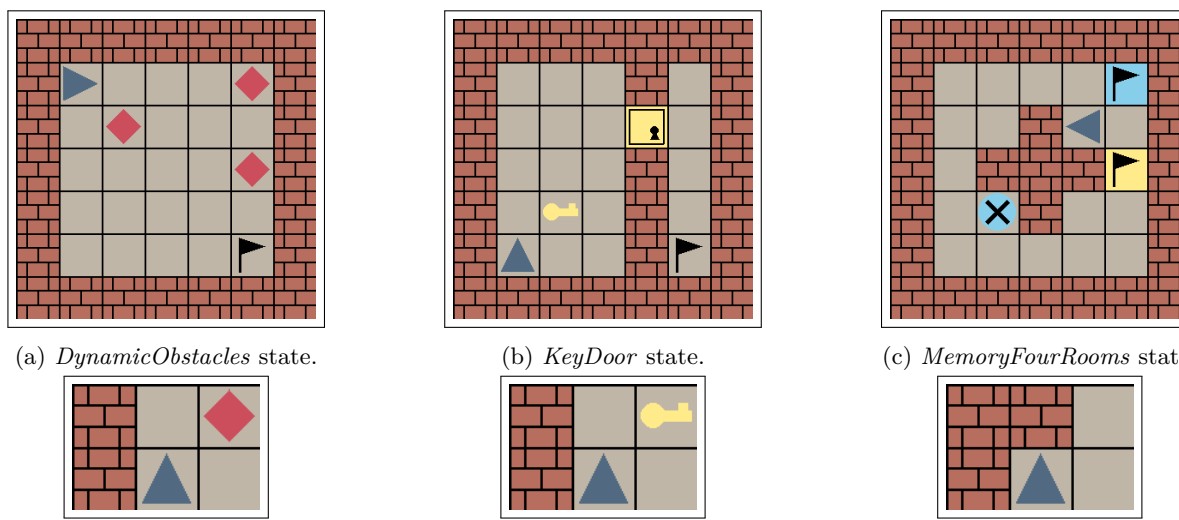

(a) *DynamicObstacles* state.      (b) *KeyDoor* state.      (c) *MemoryFourRooms* state.

(d) *DynamicObstacles* obs.      (e) *KeyDoor* obs.      (f) *MemoryFourRooms* obs.

Figure 7: *GridVerse* environments.

**Item** The item field is a 3D array representing the item (if any) that is being carried by the agent. The item's encoding is the same as in the grid field, i.e., object type, object status, and object color. In the three environments we employ, this component is not strictly necessary (not even for *KeyDoor*), so we ignore this field.

The observations of *GridVerse* environments are represented by similar components:

**Grid.** The grid field is a $2 \times 3 \times 3$ tensor of categorical data representing three channels of information for each of the $2 \times 3$ tiles observable by the agent. The channels encode the same information as in the corresponding state field.

**Agent ID Grid.** The agent ID grid field is a $2 \times 3$ bianry matrix representing the agent's location within its observation area.

**Item.** This is the same field as in the state. As in the case of the state, we ignore this field.

The actions of *GridVerse* environments vary by environment. All share common movement actions that allow the agent to step *forward*, *backward*, *left*, and *right*, as well as to turn *left* and *right*. In *KeyDoor*, additional actions allow the agent to *pick up/drop down* the key and *activate* the door.

### A.3.6 *DynamicObstacles*

In *DynamicObstacles*, the agent must reach a goal while avoiding moving obstacles. Figures 7a and 7d respectively show a sample state and observation. Agent and goal positions are fixed on two opposite corners of the grid, whereas the locations and movements of the obstacles are stochastic. In the fully-observable version of this task, the agent has full view of the grid and the obstacles, including those behind, and avoiding the obstacles is easier than in the partially-observable version of this task with limited field of view.

The agent receives a composite reward obtained by adding the following components:

- A dense living reward of $-0.05$;
- A dense reward of $+0.2$ for moving closer to the goal;
- A dense reward of $-0.2$ for moving further away from the goal;

- A sparse reward of $-1$ for bumping into a wall or an obstacle; and

- A sparse reward of $+5$ for reaching the goal.

An episode terminates upon bumping into a wall, an observable, or reaching the goal.

### A.3.7  *KeyDoor*

In *KeyDoor*, the agent must reach a goal located behind a locked door; to do so, the agent must first complete the subtasks of locating and picking up the key and opening the door. Figures 7b and 7e respectively show a sample state and observation. Positions of key, door, and goal are randomized, as well as the size of both rooms. In the fully-observable version of this task, the agent has full view of the grid, key, door, and goal. Since we ignore the *item* field of both states and observations, the fully-observable agent must infer whether it is holding the key based on whether it is observable in the grid or not, whereas the partially-observable agent must remember whether it has picked up the key in the past.

The agent receives a composite reward obtained by adding the following components:

- A dense living reward of $-0.05$;

- A dense reward of $+0.2$ for moving closer to the goal;

- A dense reward of $-0.2$ for moving further away from the goal;

- A sparse reward of $+1$ for picking up the key;

- A sparse reward of $-1$ for dropping the key;

- A sparse reward of $+1$ for opening the door;

- A sparse reward of $-1$ for closing the door; and

- A sparse reward of $+5$ for reaching the goal.

An episode terminates upon reaching the goal.

### A.3.8  *MemoryFourRooms*

In *MemoryFourRooms*, the agent must reach a good exit while avoiding a bad exit; to identify which exit is which, the agent must first explore the grid until it finds a beacon whose color matches the good exit. Figures 7c and 7f respectively show a sample state and observation. The layout of the four rooms are randomized, as well as the positions of agent, beacon, and exits. The colors of the the exits and beacon are randomly sampled from the set {red, green, blue, yellow}. In the fully-observable version of this task, the agent has full view of the grid and is able to immediately identify the good exit's identity and location, whereas in the partially-observable version, the agent must first find explore the grid to find both beacon and exit.

The agent receives a composite reward obtained by adding the following components:

- A dense living reward of $-0.05$;

- A sparse reward of $+5$ for reaching the good exit; and

- A sparse reward of $-5$ for reaching the bad exit.

An episode terminates upon reaching either exit.

## A.4  Architectures

Each environment provides state and information data in a variety of formats. This requires us to implement several different model architectures to handle these formats, the specifics of which are detailed below. Most architectures rely on an overall model that is sequentially composed of a model to learn observation representations (the nature of which depends on the environment), followed by a single recurrent layer, followed by two fully-connected (FC) layers. In each case, only observation-histories are considered, and the action-component of the history is ignored.

### A.4.1  *HeavenHell*

*HeavenHell* provides states and observations as categorical scalars. We therefore use an embedding layer to generate encodings of those values before processing with the rest of the network.

**Actor**  Embedding(256), LSTM(256), FC(256), ReLU, FC(dim($\mathcal{A}$)), Softmax

**Critic**  Embedding(256), LSTM(256), FC(256), ReLU, FC(dim($\mathcal{A}$))

### A.4.2  *CartPole/LunarLander*

*CartPole* and *LunarLander* provide states and observations as 1D feature vectors. We use a fully-connected layer to process these vectors.

**Actor**  FC(64), ReLU, LSTM(64), FC(64), ReLU, FC(dim($\mathcal{A}$)), Softmax

**Critic**  FC(64), ReLU, LSTM(64), FC(64), ReLU, FC(dim($\mathcal{A}$))

### A.4.3  *HalfCheetah*

*HalfCheetah* provides states and observations as 1D feature vectors. We use a fully-connected layer to process these vectors. The policy network outputs both action means and log standard deviations. The critic network computes action-values by combining observations and actions in its inputs and mapping them to a scalar output.

**Actor**  FC(64), ReLU, LSTM(64), FC(64), ReLU, FC($2 \cdot \dim(\mathcal{A})$)

**Critic**  FC(64), ReLU, LSTM(64), FC(64), ReLU, FC(1)

### A.4.4  *GridVerse*

*GridVerse* provides states and observations as dictionaries containing data in various formats, some categorical, some integral, and some continuous. Due to this additional complexity, we show the architecture used for *GridVerse* environments as graphical diagrams in Figure 8.

## A.5  Expert Policies

In this section, we briefly explain how we got expert policies used in our methods and evaluations, and to fill our datasets. A more in depth-discussion into the dataset composition is available in Appendix A.6.

**Fully-Observable Expert Policies.**   A key factor in our work is that fully-observable policies are usually significantly easier to train than partially-observable policies, and that we may be able to exploit fully-observable guidance to help the training of partially-observable policies. In principle, it is possible to train such fully-observable policies directly using standard offline methods on the same datasets that would be used for the partially-observable policy. However, in practice, to simplify this component and have stronger

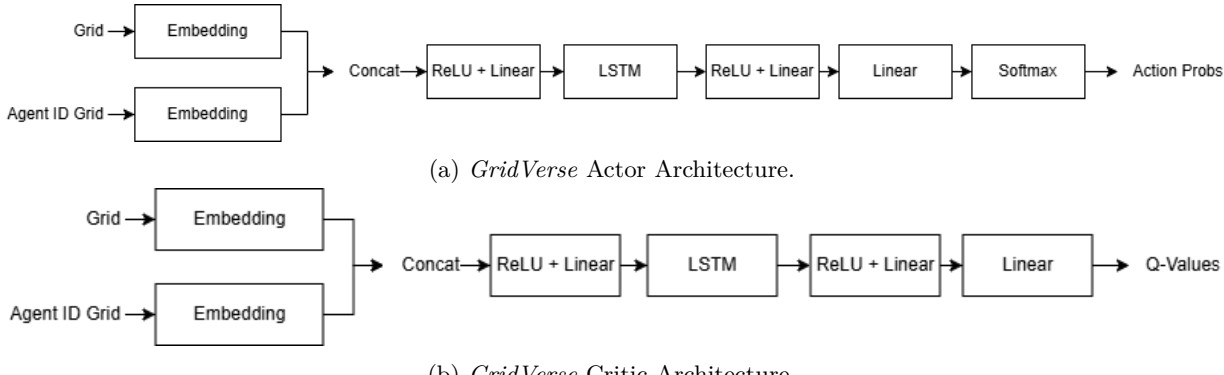

(a) *GridVerse* Actor Architecture.

(b) *GridVerse* Critic Architecture.

Figure 8: *GridVerse* Architectures.

Table 2: Datasets composition and sizes, and CO-CQL $\beta$ Coefficients

| Environment | Dataset Composition | Dataset Size | $\beta$ |
|---|---|---|---|
| Hard *HeavenHell* | (Policy) 100 % Random | 1 k episodes | 0.0001 |
| Easy *HeavenHell* | (Policy) 100 % Expert | 100 k timesteps | 0.001 |
| Easy *HeavenHell* | (Policy) 50 % Expert, 50 % Random | 100 k timesteps | 0.01 |
| Easy *HeavenHell* | (Policy) 100 % Random | 100 k timesteps | 0.0001 |
| Latent *CartPole* | (Policy) 100 % Expert | 1 M timesteps | 0.02 |
| Latent *CartPole* | (Policy) 50 % Expert, 50 % Random | 1 M timesteps | 0.02 |
| Latent *CartPole* | (Policy) 100 % Random | 1 M timesteps | 1.0 |
| Noisy *LunarLander* | (Policy) 100 % Trained | 1 k episodes | 0.9 |
| Noisy *HalfCheetah* | (Policy) 100 % Trained | 1 k episodes | 0.5 |
| Latent *LunarLander* | (Policy) 100 % Trained | 1 k episodes | 0.0001 |
| Latent *HalfCheetah* | (Policy) 100 % Trained | 1 k episodes | 0.4 |
| *MemoryFourRooms* | (Trajectories) 50 % Success, 50 % Failure | 1 k episodes | 0.1 |
| *KeyDoor* | (Trajectories) 50 % Success, 50 % Failure | 1 k episodes | 0.1 |
| *DynamicObstacles* | (Trajectories) 50 % Success, 50 % Failure | 1 k episodes | 0.1 |

optimality guarantees on the fully-observable expert, we employ online RL methods. To obtain the fully-observable experts for *HeavenHell*, *CartPole*, *LunarLander*, and *HalfCheetah*, we employ standard non-recurrent SAC (Haarnoja et al., 2018). To obtain the fully-observable experts for *GridVerse*, we employ standard non-recurrent PPO (Schulman et al., 2017).

It should be additionally noted that fully-observable experts can also be obtained, depending on the setting, without requiring any training at all, e.g. a coded heuristic could act as an expert policy, or behavior cloning onto a human expert.

**Partially-Observable Expert Policies.** To create some of the datasets used in our evaluation, we will need a way to generate expert partially-observable trajectories. To generate those trajectories, we train partially-observable policies using recurrent SAC (Haarnoja et al., 2018).

### A.6 Datasets

The composition and size of each dataset varies per environment, as shown in Table 2. In the table, we use the following labels:

**(Policy) Random.** This label denotes trajectories obtained by running a random policy.

**(Policy) Expert.** This label denotes trajectories obtained by running a partially-observable policy trained as described in Appendix A.5. Further, we were able to verify that the task is being solved well, e.g., by meeting a known return threshold that is considered to be significant for the task.

**(Policy) Trained.** This label also denotes trajectories obtained by running a partially-observable policy trained as described in Appendix A.5. However, in this case we were not able to verify that the task is being solved well, so the policy is still sub-optimal in some regard.

**(Trajectory) Success.** This label denotes trajectories that are sampled from a random policy, but also filtered so that the correct goal is reached. The filtering process does not account for whether the task was solved efficiently or even in a manner that is congruent to optimal partially-observable control, e.g., a trajectory is deemed successful even if it reaches a *good* exit without having made a corresponding observation necessary to identify it from a *bad* exit.

**(Trajectory) Failure.** This label denotes trajectories that are sampled from a random policy, but also filtered so that the correct goal is *not* reached.

All datasets will be made publicly available upon publication.

### A.7 Hyperparameters

In this section, we show in details the hyperparameters used to train CO-CQL and the other baselines.

### A.7.1 CO-CQL Hyperparameters

The general hyperparameters used for CO-CQL for training across all of our datasets are shown in Table 3. The specific $\beta$ coefficient used with each of our datasets is shown in Table 2.

We tune primarily the behavior cloning coefficient $\beta$ and the CQL alpha threshold value. All other hyperparameters are carried over un-modified from standard online, non-recurrent SAC as well as non-recurrent CQL (besides batch size and history length which we choose based on computational capacity). We performed simple grid-searching to determine our $\beta$ coefficients.

We find from the COAR analysis in section 5.2.5 that if one has intuition regarding the MDP/POMDP overlap of an environment, then the search space over the optimal $\beta$ parameter can be shortened; if the expected overlap is higher, then one should start with higher values of $\beta$ and vice versa.

### A.7.2 CQL Hyperparameters

The general hyperparameters used for recurrent CQL for training across all of our datasets are shown in Table 3.

### A.7.3 IQL Hyperparameters

The general hyperparameters used for recurrent IQL for training across all of our datasets are shown in Table 5.

### A.7.4 BC Hyperparameters

The general hyperparameters used for recurrent BC for training across all of our datasets are shown in Table 6.

### A.7.5 TD3+BC Hyperparameters

The general hyperparameters used for recurrent TD3+BC for training across all of our datasets are shown in Table 7.

Table 3: CO-CQL Hyperparameters

| Hyperparameter | Environment | Value |
|:---:|:---:|:---:|
| Discount $\gamma$ | all | 0.99 |
| Batch Size | all | 32 |
| History Length | *HeavenHell* *GridVerse* | Full episodes |
| | *CartPole* *LunarLander* *HalfCheetah* | 4 timesteps |
| Actor Learning Rate | *HeavenHell* *GridVerse* | $3e-5$ |
| | *CartPole* *LunarLander* *HalfCheetah* | $3e-4$ |
| Critic Learning Rate | all | $3e-4$ |
| Actor Update Frequency | all | 2 |
| Target Network Update Frequency | all | 1 |
| Target Entropy Scaling | all | 0.3 |
| CQL $\tau$ | Noisy *HalfCheetah* | 2 |
| | Latent *HalfCheetah* Latent *LunarLander* Noisy *LunarLander* | 1 |
| | *HeavenHell* *GridVerse* *CartPole* | 10 |

Table 4: CQL Hyperparameters

| Hyperparameter | Environment | Value |
|---|---|---|
| Discount $\gamma$ | all | 0.99 |
| Batch Size | all | 32 |
| History Length | *HeavenHell* *GridVerse* | Full episodes |
| | *CartPole* *LunarLander* *HalfCheetah* | 4 timesteps |
| Actor Learning Rate | *HeavenHell* *GridVerse* | $3e-5$ |
| | *CartPole* *LunarLander* *HalfCheetah* | $3e-4$ |
| Critic Learning Rate | all | $3e-4$ |
| Actor Update Frequency | all | 2 |
| Target Network Update Frequency | all | 1 |
| Target Entropy Scaling | all | 0.3 |
| CQL $\tau$ | Noisy *HalfCheetah* | 2 |
| | Latent *HalfCheetah* Latent *LunarLander* Noisy *LunarLander* | 1 |
| | *HeavenHell* *GridVerse* *CartPole* | 10 |

Table 5: IQL Hyperparameters

| Hyperparameter | Environment | Value |
|---|---|---|
| Discount $\gamma$ | all | 0.99 |
| Batch Size | all | 32 |
| History Length | *HeavenHell GridVerse* | Full episodes |
| | *CartPole LunarLander HalfCheetah* | 4 timesteps |
| Actor Learning Rate | all | 3e−4 |
| Critic Learning Rate | all | 3e−4 |
| State-Value Function Learning Rate | all | 3e−4 |
| Actor Update Frequency | all | 2 |
| Target Network Update Frequency | all | 1 |
| IQL $\tau$ | all | 0.7 |
| IQL $\beta$ | all | 3.0 |

Table 6: BC Hyperparameters

| Hyperparameter | Environment | Value |
|---|---|---|
| Discount $\gamma$ | all | 0.99 |
| Batch Size | all | 32 |
| History Length | *HeavenHell GridVerse* | Full episodes |
| | *CartPole LunarLander HalfCheetah* | 4 timesteps |
| Actor Learning Rate | all | 3e−4 |
| Actor Update Frequency | all | 2 |

Table 7: TD3+BC Hyperparameters

| Hyperparameter | Environment | Value |
|---|---|---|
| Discount $\gamma$ | *HalfCheetah, LunarLander* | 0.99 |
| Batch Size | *HalfCheetah, LunarLander* | 32 |
| History Length | *HalfCheetah, LunarLander* | 4 timesteps |
| Actor Learning Rate | *HalfCheetah, LunarLander* | 3e−4 |
| Critic Learning Rate | *HalfCheetah, LunarLander* | 3e−4 |
| Actor Update Frequency | *HalfCheetah, LunarLander* | 2 |
| Target Network Update Frequency | *HalfCheetah, LunarLander* | 1 |
| TD3+BC $\alpha$ | *HalfCheetah, LunarLander* | 2.5 |
| Exploration noise | *HalfCheetah, LunarLander* | 0.1 |
| Policy noise | *HalfCheetah, LunarLander* | 0.2 |
| Noise clip | *HalfCheetah, LunarLander* | 0.5 |

