# OpenReview forum: "Leveraging Fully-Observable Solutions for Improved Partially-Observable Offline Reinforcement Learning"
_TMLR — Accepted by TMLR_

### Review · Reviewer_M5qG · 2025-06-05

**Summary Of Contributions:**

This paper proposes a method that leverages fully-observable expert policies to partially-observable control settings. Particularly, it incorporates historical data into the network input to mitigate the influence of partial observability.

**Audience:**

Yes

**Broader Impact Concerns:**

None.

**Claims And Evidence:**

No

**Requested Changes:**

* What does *hao* mean in $Q(hao, a')$ in Equation 11? It also appears in other equations.

**Strengths And Weaknesses:**

**Strengths**

* The problem studied in this paper is well-motivated.

* The presentation is clear and easy to follow.

* Empirical results are analyzed in detail. I am particularly interested in the latent state dimension example, as we usually do not have access to the higher-order information such as velocity in many control problems.

**Weaknesses**

* In the abstract, it claims that the proposed CO-CQL algorithm leverages fully-observable expert policies in an asymmetric learning setting, without mentioning the need of a well-trained partially-observable policy $\pi$ which appears in Definition 4.2 and serves as a keystone in CO-CQL. I wonder if the proposed approach still works when there is only a well-trained fully-observable policy $\mu$ available as in the usual setting of offline RL.

* One of the core contributions of this paper is the proposed COAR metric which serves as a practical estimate of COOR. However, it does not contribute to the proposed CO-CQL algorithm at all. While I understand that it still has its strength in evaluating the post-training performance, its importance is not as clear as suggested.

* The proposed algorithm is compared only with other baseline methods that are not supposed to handle the partially-observable problems. A more convincing way to demonstrate its strength is to compete with approaches that are particularly designed for this setting. An example can be found in the paper by Morad et al. (https://arxiv.org/pdf/2303.01859).

* There is a lack of concrete approach to transition from partially-observable data to sequential data suggested in Section 4.2 (namely, from $\pi(s)$ to $\pi(h)$).

* The selection of hyperparameter can be critical to the performance of the proposed method as shown in Figure 4 and in Table 1. Can we get a better understanding towards it than those elaborated in section 5.2.5?

* Experiments are rather simple. Since the main contribution of this paper is on the empirical side, it would be more convincing if more difficult experiments (e.g., hopper, walker and humanoid) can be conducted and presented.

Overall, this paper needs a major revision to clarify its contributions and provide more convincing evidence.

---

### Review · Reviewer_5Y9P · 2025-06-30

**Summary Of Contributions:**

- Work introduces fresh take on offline RL for PO control using FO expert policies without needing online environment interactions. This makes it practical for cases like self driving or robotics where richer data will be there during training but not on deployment
- new theoretical measure called  Cross-Observability Optimality Ratio (COOR), a nice way to formalize the intuition that FO guidance can help PO tasks.
- Propose emperical way to compute COOR by defining COAR (Cross Observability Approximation Ratio) as a workable alternative to COOR
- Proposes new algorithm (Cross-Observable Conservative Q-Learning) that mixes conservative Q-learning with a behavior cloning loss to learn from FO experts
- Tested proposed CO-CQL on a bunch of different PO control tasks
- Robustness analysis via experiements for dataset variations
- Have some useful guidance for practitioners like their observation that tasks where FO and PO actions align a lot, even simple behavior cloning can do well and that CO-CQL can push that performance further

**Audience:**

Yes

**Broader Impact Concerns:**

I don't see any significant ethical implications in this work as it focuses on technical advancements in offline RL and from my reading I don't see social concerns either

**Claims And Evidence:**

Yes

**Requested Changes:**

- The paper does not clearly articulate how CO-CQL advances (fundamentally) beyond existing methods. e.g., the addition of behavior cloning is extension of TD3+BC (as given in paper). Why is this proposed hybrid approach needed when BC alone performs well in high COOR env. ? A stronger justification is needed.

- Also as a continuation, if the authors can clearly articulate how C0-CQL advances beyond TD3+BC, CQL ... may be by demostrating unique scenarios where C0-CQL outperforms simpler BC or standard CQL in low-COOR environments, that would strengthen this work

- COOR seems like a motivational tool rather than a component in C0-CQL and COAR as an approximation that relies on well-trained policies and probability threshold. How is this threshold chosen? how is COAR computed in practise?  The empirical validation of COAR's correlation with BC performance will strengthen the robustness of this approach

- The HeavenHell example oversimplifies the problem, as it assumes a specific scenario where FO and PO actions overlap significantly. How does the proposed COOR and COAR generalize to more complex PO tasks with less overlap?

- Also analysis of the same example is not convincing: the FO expert ignores the oracle, yet CO-CQL learns to visit it? The work says, "CO-CQL learns to exploit the fully-observable guidance of how to reach good exit, while learning key piece of info." How does the policy network arbitrate between these conflicting objectives? A deeper analysis here is required, without it it reads more accidental than by design

- The comparison against baselines (e.g., CQL, IQL, BC) is not rigorous. e.g.,, the paper claims CO-CQL outperforms other methods but Table 2 only lists dataset compositions and \beta without performance metrics. The lack of quantitative results (e.g., average returns, success rates) makes it impossible to verify the claims proposed in the paper.

- For baselines, the discussion should more directly address why cloning a potentially OOD FO expert is superior to cloning the in-distribution behavior policy? A more insightful baseline (something like: where BC loss clones the dataset actions instead of the FO expert's actions?) would strengthen this work

- The dataset compositions are varied, but the paper does not discuss how these variations impact CO-CQL’s performance relative to baselines. Why not compare CO-CQL against a purely BC approach in high-COOR environments, given the paper’s admission that BC performs well in such cases?

- The paper assumes access to a dataset containing both states and observations, which is a strong assumption for offline PO RL. In many real-world scenarios (at least from my experience in process manufacturing, ads and healthcare), only observation data may be available during deployment. The work briefly mentions scenarios where state information is accessible during training like simulation, but it does not address how CO-CQL would perform with observation-only datasets. Could you add more details around it? because without it, it would limit the method’s applicability to truly PO settings.

- Include a theoretical analysis of CO-CQL’s convergence or robustness, even if simplified as this would provide confidence in its performance to readers.


Minor comments:

- The dataset compositions in Table 2 include mixtures of expert, random, and trained policies, but the process for generating expert policies is only briefly described. The paper states that experts are trained using the same dataset, but no details are provided on the training algorithm. More info here will be helpful

- The paper briefly mentions that representation learning techniques could be integrated with CO-CQL in future work. However, it does not explain why these techniques were not considered in the current evaluation, given their relevance to PO RL. A comparison to such methods would have strengthened the paper’s positioning in the literature.

- The \beta coefficients in Table 2 vary very widely (from 0.001 to 1.0), but no explanation/analysis is given on how these diverse ranges or  these hyperparameters would have on performance.

**Strengths And Weaknesses:**

Strengths:

- Tackles under explored problem - RL for PO control
- Problem formulation via asymmetric learning framework where agents can lean on richer state info during training  before deploying in the real world
- Introduction of COOR and COAR concepts, a fresh way to think about the problem with how much FO and PO optimal actions overlap
- Broad testing across diverse tasks and consideration of dataset quality variations

Weaknesses:

- Not a big leap in Novelty - the work builds on existing work like COSIL &, CQL
- COOR and COAR feel disconnected, these two concepts seem more of motivational than practical as they are not directly used in proposed CO-CQL
- Emperical results are thin as the provided results does not have performance numbers
- The work assumes having a dataset with both states and observations during training, which is a big ask for truly PO setting
- No theoritcal guarantees about CO-CQL behavior

---

> ### Author Response · Authors · 2025-07-15
> **Response to reviewer, part 1**
>
> We thank the reviewer for taking the time to provide their thoughtful feedback. Below, in the following series of comments, we address the main concerns raised:
>
> **Empirical results are thin as the provided results does not have performance numbers; The comparison against baselines (e.g., CQL, IQL, BC) is not rigorous…**
>
> We thank the reviewer for the comment, but respectfully disagree. Figure 2 provides quantitative learning curves across eight domains and demonstrates CO-CQL’s robust performance across all of the domains against multiple baselines. Figure 4 provides learning curves for our analysis of the behavior cloning coefficient beta. Figure 5 provides learning curves for our dataset analysis results. In addition, the learning curves demonstrate not just the final performance, but also other aspects like convergence speed. If the reviewer is able to provide more specificity regarding what they would like that isn’t covered by our existing results, that would be greatly appreciated!
>
> **If the authors can clearly articulate how CO-CQL advances beyond TD3+BC, CQL — maybe by demonstrating unique scenarios where CO-CQL…**
>
> We thank the reviewer for raising the points about novelty and would like to clarify.
>
> Offline partial observability introduces unique challenges that are not handled by the existing methods. TD3+BC and CQL are strong baselines in fully-observable offline RL, but they are not designed to cope with partial observability, particularly when behavior cloning from a fully-observable expert can lead to incorrect actions due to missing information. Conversely, COSIL is an online method that exploits asymmetric supervision to accelerate exploration, whereas CO-CQL operates in the fully offline setting where no environment interaction (and no explicit exploration) is possible.
>
> We note that intermediate COOR examples like HeavenHell are exactly the scenarios that demonstrate the utility of CO-CQL. If COOR were much higher, then pure BC would be better suited (as there is no need to deviate from the expert policy), and if COOR were much lower, then pure CQL would be better suited (as there is nothing to be learned from the expert policy). Ultimately, there is no free lunch, and different methods are better tailored for different situations. Nonetheless, our results in Figure 2 demonstrate that CO-CQL is still able to outperform both pure BC as well as CQL in a robust set of varying COOR environments.
>
> **COOR seems like a motivational tool rather than a component in CO-CQL, and COAR as an approximation…**
>
> Neither COOR nor COAR are explicitly used during training; they are indeed motivational and analytical tools. Its role is to formalize the intuition that some partially-observable tasks share action overlap with their fully-observable counterparts, and thus benefit from expert supervision, while others do not. COAR serves as a practical, post-training approximation of COOR. As explained in section 4.1, it is computed by comparing the final overlap between high-probability actions chosen by the expert and the trained policy. In practice, we use threshold q=0.1, and found COAR estimates to be robust across nearby values. We will clarify this in the main text. Section 5.2.5 and Table 1, show that COAR correlates well with the best-performing BC coefficient \beta, which supports its utility as a diagnostic tool for analyzing the effectiveness of asymmetric supervision.
>
> **The HeavenHell example oversimplifies the problem…**
>
> We appreciate this concern, however we at least partially disagree that HeavenHell is an oversimplification. HeavenHell represents a core archetype of partially observable tasks that require the agent to first perform information-gathering and then to act upon the information. Other archetypes indeed exist, with different observation and information-gathering structures, but the structure of HeavenHell represent an important crucial form of partial observability.
>
> **Also analysis of the same example is not convincing: the FO expert ignores the oracle, yet CO-CQL learns to visit it?**
>
> We appreciate the reviewer’s request for a deeper explanation. While the FO expert ignores the oracle, the agent as a whole is still able to learn from the interactions available in the offline dataset, which do visit the oracle. Overall, visiting the oracle is easy and can be done despite going against the expert’s guidance, but learning to associate the oracle’s information with the correct exit is much harder, and that is where the expert’s guidance comes to full fruition. As a total net effect, the advantage is much stronger than the disadvantage, hence the agent learns to extrapolate from the expert’s guidance and solve the task completely.

---

> > ### Author Response · Authors · 2025-07-15
> > **Response to reviewer, part 2**
> >
> > **The discussion should more directly address why cloning a potentially OOD FO expert is superior to cloning the in-distribution behavior policy…**
> >
> > We thank the reviewer for this suggestion and are happy to clarify our reasoning. The primary reason to avoid cloning from the in-distribution behavior is to avoid making any assumption about the quality of the in-distribution actions, and allow to apply CO-CQL to datasets that contain purely exploratory or other suboptimal behaviors. It is clear that cloning such actions should provide no benefit. We do additionally note (as in appendix A.3), that the fully observable expert can be in practice obtained by training an offline RL algorithm on the same dataset.
> >
> > **The dataset compositions are varied, but the paper does not discuss how these variations impact CO-CQL’s performance relative to baselines…**
> >
> > We are happy to clarify this point. Our evaluation includes multiple environments and dataset variants specifically to study how CO-CQL performs under different levels of expert signal and partial observability, including high-COOR settings where behavior cloning (BC) performs well.
> > We do in fact include pure BC baselines in all experiments, including high-COOR environments such as Noisy LunarLander and KeyDoor (see Figure 2e and 2h). In these settings, BC achieves strong performance, and our results show that CO-CQL matches or slightly exceeds BC, indicating that CO-CQL does not degrade performance in high-overlap cases, while still providing benefits in more challenging environments.
> > Regarding dataset composition: Section 5.2.6 (Figure 5) directly analyzes how varying the proportion of expert vs. non-expert trajectories in the dataset affects performance. We show that CO-CQL remains robust across different dataset mixtures, while baselines like IQL and CQL degrade more sharply when dataset quality decreases.
> >
> > **The paper assumes access to a dataset containing both states and observations, which is a strong assumption for offline PO RL…**
> >
> > We appreciate the reviewer’s point, and would like to clarify two points. CO-CQL learns a policy that only requires observations during deployment and thus is fully compatible with execution in partially-observable environments. The reviewer is correct that CO-CQL cannot be used if state information is not available in the training data. However, there are many scenarios where that state information can be available. Most notably, if the dataset is collected by running a simulator, but also more broadly possible to collect training data that includes additional sensors compared to the deployment sensors (e.g., training in a controlled warehouse with many cameras, for execution in the open or in a generic household environment).
> >
> > **No theoretical guarantees about CO-CQL behavior; Include a theoretical analysis of CO-CQL’s convergence or robustness…**
> >
> > We thank the reviewer for raising this point. Our work is indeed primarily empirical in nature, however, CO-CQL has convergence guarantees similar to other prior methods: the value model converges following the same arguments as for CQL, and the policy converges due to being optimized to minimize a well-defined objective via SGD.
> >
> > **The dataset compositions in Table 2 include mixtures of expert, random, and trained policies…**
> >
> > We appreciate the reviewer’s comment. Appendix A.3 provides the full details of how we generate both our fully-observable and partially-observable expert policies. As mentioned, to obtain the fully observable experts for HeavenHell, CartPole, LunarLander, and HalfCheetah, we employ standard non-recurrent SAC. To obtain the fully-observable experts for GridVerse, we employ standard non-recurrent PPO. Our partially-observable expert policies were trained using recurrent SAC.
> >
> > **The paper briefly mentions that representation learning techniques could be integrated with CO-CQL…**
> >
> > We appreciate the reviewer’s insight and agree that comparing to representation learning techniques is a valuable direction for future work. In this paper, we intentionally focused on evaluating the core contribution of CO-CQL: leveraging fully-observable expert policies for partially-observable offline RL. This required isolating the impact of asymmetric supervision (across different types of partial observability and dataset quality) to rigorously assess when and how it provides benefits.
> > Integrating representation learning methods (e.g., contrastive learning, recurrent encoders with auxiliary objectives) introduces additional variables which could confound the interpretation of CO-CQL’s performance gains. Our goal was to evaluate CO-CQL in a clean and controlled setting.

---

### Review · Reviewer_AWnz · 2025-07-04

**Summary Of Contributions:**

This paper introduces a new way to teach AI agents to make decisions when they can only partially see their environment, using pre-recorded data rather than live practice. The key innovation is a method called CO-CQL that learns from two sources: the pre-recorded data and an "expert" AI that has full visibility of the environment. The researchers first developed mathematical tools (COOR and COAR) to measure when it's helpful to learn from the fully-visible expert. They then created CO-CQL, which combines learning from the recorded data with copying the expert's behavior when appropriate. They tested their method on various challenges, like when the AI has noisy sensors or limited vision, and found it performed better than existing approaches. The method is particularly clever because it can figure out when to trust the expert's advice and when to ignore it. This research is important because most current methods assume AI agents can see everything in their environment, which isn't realistic for many real-world applications like robotics or self-driving cars.

**Audience:**

Yes

**Broader Impact Concerns:**

No significant ethical concerns

**Claims And Evidence:**

Yes

**Requested Changes:**

1. Hyperparameter Analysis

- Add basic sensitivity analysis for behavior cloning coefficient β
- Provide clearer guidelines for selecting β across different environments
- This is critical as β is a key parameter affecting performance and current guidance is limited

2. Theoretical Analysis

- Add basic convergence analysis for CO-CQL
- Explore theoretical relationship between COOR and performance
- Would strengthen theoretical foundations but not essential for practical utility

**Strengths And Weaknesses:**

Strengths -

1. New Theoretical Contribution
- Well-defined metrics (COOR and COAR) for quantifying when fully-observable experts can help
- Clear mathematical framework that bridges theory and practice

2. Practical Algorithm Development
- CO-CQL effectively combines behavior cloning with offline RL
- Successfully handles both discrete and continuous action spaces
- Works across diverse partially-observable scenarios

3. Strong Empirical Validation
- Comprehensive testing across different types of partial observability
- Clear performance improvements over baselines
- Good analysis of robustness to dataset quality

Weaknesses -

1. Limited Theoretical Analysis
- Lacks formal convergence guarantees
- Could benefit from theoretical bounds on performance

2. Hyperparameter Sensitivity
- Context-independent behavior cloning coefficient (β) despite context-dependent optimal values
- Limited analysis of sensitivity to other hyperparameters

3. Practical Considerations
- Could provide more guidance on expert policy selection
- Limited discussion of computational requirements and scalability
- Missing analysis of sample efficiency

Overall, the strengths outweigh the limitations, and the weaknesses are resolvable through additional analysis

---

> ### Author Response · Authors · 2025-07-15
> **Response to reviewer**
>
> **Theoretical Analysis/Limited Theoretical Analysis**
>
> We thank the reviewer for raising this point. Our work is indeed primarily empirical in nature, however, CO-CQL has convergence guarantees similar to other prior methods: the value model converges following the same arguments as for CQL, and the policy converges due to being optimized to minimize a well-defined objective via SGD.
>
> **Practical Considerations**
>
> We thank the reviewer for bringing up additional points about practicality. Appendix A.3 contains a full breakdown of the MDP/POMDP policies, their motivations, and how they were generated. A comprehensive analysis on scalability and sample-efficiency is near-impossible to provide, as these aspects are strongly driven by the properties of individual environments and modeling choices, over the learning algorithm itself. Because of these difficulties, as is common in the broad RL literature, we focus on an empirical evaluation to demonstrate the validity of asymmetric learning in the offline RL setting.
>
> **Hyperparameter Sensitivity/Hyperparameter Analysis**
>
> We thank the reviewer for highlighting points about hyperparameter analysis and sensitivity. As noted in Section 5.2.5 and Appendix A.5, all hyperparameters are kept fixed across runs, except for the behavior cloning coefficient β. This was a deliberate choice to isolate the effect of β and allow a clean analysis of its impact, shown in Figure 4 and Table 1. Our results also demonstrate that CO-CQL is robust across a broad range of β values, particularly in high-COAR environments.
> With regards to the behavior cloning coefficient β, we include a detailed sensitivity analysis in Section 5.2.5, which examines CO-CQL performance across a range of beta values on multiple environments with varying levels of cross-observability, and Figure 4 displays the associated learning curves. The results in that section support the general conclusion that if one expects an environment to have higher overlap between its MDP and POMDP versions then a higher beta value is likely good to start with, whereas if a lower overlap is expected then a lower beta value is better to start with. We have updated Appendix A.5.1 that discusses CO-CQL hyperparameters with these additional conclusions.

---

### Decision · Action_Editor_ehAC · 2025-08-07

**Recommendation:** Accept with minor revision

**Additional Comments:**

The paper shows an innovative approach to improve offline RL in partially-observable environments, and introduces theoretical metrices to quantify this benefit.

After extensive adjustments based on the reviews the resulting improvements, the paper can be accepted.

However, a revision is still necessary:

The HeavenHell arbitration mechanism and theoretical guarantees need to be worked out in more detail in order to be completely convincing. If possible a case study with low COOR, an experiment for observation only, a paragraph on compensating for the loss function, a scenario for observation only, and a more in-depth HeavenHell analysis should be added.

**Audience:**

Yes

**Audience Explanation:**

The reviewers agree that the paper is interesting. In particular, it is emphasized that it shows an innovative approach of leveraging fully-observable expert solutions to improve offline RL in partially-observable environments, and the introduction of the theoretical metrices COOR and COAR to quantify this benefit.

**Claims And Evidence:**

Yes

**Claims Explanation:**

Most of the criticisms regarding the lack of evidence were addressed in the authors' responses to the reviews.

However, according to Reviewer 5Y9P, there are still some weaknesses that need to be addressed:

The HeavenHell arbitration mechanism and theoretical guarantees need to be worked out in more detail in order to be completely convincing. The gaps can be closed with minor changes—by adding a case study with low COOR, an experiment for observation only, a paragraph on compensating for the loss function, a scenario for observation only, and a more in-depth HeavenHell analysis.